# Corrosion–Resistance Mechanism of TC4 Titanium Alloy under Different Stress-Loading Conditions

**DOI:** 10.3390/ma15134381

**Published:** 2022-06-21

**Authors:** Xin-Yu Wang, Shi-Dong Zhu, Zhi-Gang Yang, Cheng-Da Wang, Ning Wang, Yong-Qiang Zhang, Feng-Ling Yu

**Affiliations:** 1School of Materials Science and Engineering, Xi’an University of Petroleum, Xi’an 710065, China; wangxy996@126.com (X.-Y.W.); mecury1001@163.com (F.-L.Y.); 2Shaanxi Key Laboratory of Carbon Dioxide Sequestration and Enhanced Oil Recovery, Shaanxi Yanchang Petroleum (Group) Co., Ltd., Xi’an 710065, China; zgyang1031@163.com (Z.-G.Y.); wangcd9999@126.com (C.-D.W.); wangning8103@163.com (N.W.); zhangyqslb@163.com (Y.-Q.Z.)

**Keywords:** TC4 titanium alloy, stress load, electrochemical test, pitting mechanism

## Abstract

Titanium alloys have now become the first choice of tubing material used in the harsh oil- and gas-exploitation environment, while the interaction of force and medium is a serious threat to the safety and reliability of titanium alloy in service. In this paper, different stresses were applied to TC4 titanium alloy by four-point bending stress fixture, and the corrosion behavior of TC4 titanium alloy was studied by high-temperature and high-pressure simulation experiments and electrochemical techniques, and the microscopic morphologies and chemical composition of the surface film layer on the specimen were characterized by scanning electron microscopy (SEM), transmission electron microscopy (TEM), X-ray energy-dispersive spectroscopy (EDS), X-ray diffraction (XRD), and X-ray photoelectron spectroscopy (XPS), to reveal the corrosion-resistance mechanism of TC4 titanium alloy under different stress-loading conditions. The results showed that the pits appeared on the specimens loaded with elastic stress, but the degree of pitting corrosion was still lighter, and the surface film layer showed n-type semiconductor properties with cation selective permeability. While the pits on the specimens loaded with plastic stress were deeper and wider in size, and the semiconductor type of the surface film layer changed to p-type, it was easier for anions such as Cl^−^ and CO_3_^2−^ to adsorb on, destroy, and pass through the protective film and then to contact with the matrix, resulting in a decrease in corrosion resistance of TC4 titanium alloy.

## 1. Introduction

Titanium alloys will rapidly form a layer of continuous and dense oxide film on the surface after contacting with the corrosive ions, although the film layer is relatively thin, only 3–5 μm, which enables titanium alloys to exhibit high corrosion resistance under many corrosive conditions [1,2,3]. Together with excellent overall properties such as low density and high specific strength, titanium alloys are highly valuable for applications in automotive [4], marine [5], aerospace [6,7], and oil- and gas-exploitation fields [8,9,10,11], and are also widely used in biomedical fields due to their biocompatibility and good interaction with the human environment [12,13].

In oil and gas field, with the continuous development of (ultra-)deep wells, the service environment of tubing has become more and more harsh. Abroad, the vertical depth of most of wells in the Gulf of Mexico, North Sea and other beach sea, deep water, and other operating areas exceeds 9000 m, the formation temperature exceeds 200 °C, and the formation pressure exceeds 140 MPa [14]. Southwest and northwest China and other super-scale gas fields are also mostly characterized by deep burial, high temperature and pressure, and high sulfide and CO_2_ concentration, such as the partial pressure of CO_2_ of the higher-acidic gas field in northeast Sichuan, which is higher than 6 MPa, and the temperature is higher than 150 °C [15]. In addition, the depth of oil and gas wells in China’s Tarim oil field also exceeds 8000 m, and the bottom pressure is higher than 100 MPa, the temperature is higher than 180 °C, and the erosive ions such as Cl^−^ concentration in produce fluid often exceeds 1.2 × 10^5^ mg/L [16]. Cl^−^ is notorious for its small ionic radius, high penetration, and strong adsorption on metal surfaces [17,18]. As a negatively charged ion, Cl^−^ easily penetrates the corrosion product layer and/or the passive film to reach the interface between the metal matrix and the passivation film, leading to the amorphization of the metal lattice and accelerated anodic dissolution [19,20]. Cl^−^ can also form a chloride salt layer on the metal surface, replacing the protective passivation film, thus accelerating the onset of pitting corrosion [21,22]. In addition, the downhole tubular column produces stress due to the action of self-weight, well slope, temperature, and buckling. These corrosion factors either act singly or coupled on the corrosion behavior of the downhole tubular, and the passivation film on the surface of tubing undergoes local breakage, resulting in localized corrosion or uniform corrosion. In particular, pitting corrosion provides the initiation point for stress-corrosion crack (SCC) extension or corrosion fatigue, both of them further reducing the useful life of the material [23,24,25]. Sanderson [26,27] emphasized that significant SCC effects were found for Ti-6Al-4V alloy in pure methanol and methanol-HCl solutions if considering the results of u-shaped specimens. Therefore, an accurate assessment of this corrosion process is needed to mitigate damage to alloyed components [28,29].

In this paper, the corrosion behavior of TC4 (Ti-6Al-4V) titanium alloy in CO_2_-Cl^−^ systems was studied, and corrosion-resistance mechanism of TC4 titanium alloy under different stress conditions was revealed.

## 2. Materials and Methods

### 2.1. Materials

The specimens of TC4 titanium alloy used in this experiment were mainly composed of α + β phase isometric crystals, as shown in Figure 1, and their chemical composition was shown in Table 1. The organization was uniform and fine, the black phase was α phase, the white phase was β phase, β phase was distributed in grain boundary of α phase, and part of the α phase was elongated along the rolling direction to form a fixed fiber direction. The isometric organization had good plasticity and elongation, high surface shrinkage, and good resistance to notch sensitivity and thermal stability, as shown in Table 2.

According to National Association of Corrosion Engineer (NACE) standard SP0775-2013 “Preparation, Installation, Analysis, and Interpretation of Corrosion Coupons in Oilfield Operations”, TC4 titanium alloy specimens were processed into 65 mm × 10 mm × 3 mm rectangular specimens.

According to the standard GB/T-15970.2-2000 “Corrosion of metals and alloys--Stress corrosion testing—Part 2: Preparation and use of bent-beam specimens”, a four-point bending stress fixture was used to load the specimen with constant deformation, and the four-point bending loading stress was calculated as follows:(1)$$\sigma =\frac{12\mathrm{Ety}}{{3H}^{2}-{4A}^{2}}$$
where σ: maximum tensile stress, MPa;E: modulus of elasticity, MPa;t: thickness of specimen, mm;y: maximum deflection, mm;H: distance between external support points, mm;A: distance between the internal and external support points, mm.The relevant dimensions of the fixture were H = 52 mm, t = 3 mm, E = 110 × 10^3^ MPa, A = 13 mm, and the specific values are shown in Table 3, where σ_s_ corresponds to R_p0.2_ in Table 2.


Scanning electron microscopy (SEM: JSM-6390A, Nippon Electron Corporation, Minato, Japan) was used to observe the microscopic corrosion morphology on the surface of the specimens. X-ray energy-dispersive spectrometer (EDS), an additional module of SEM for composition analysis, was used to analyze the elements of corrosion products on the surface of the specimens. X-ray diffractometer (XRD-6000, Shimadzu, Kyoto, Japan), with a scanning range of 5–144° (2θ) and an X-ray tube power of 2.7 KW, was used to characterize the corrosion products on the surface of the specimen qualitatively. X-ray photoelectron spectroscopy (XPS: Thermo SCIENTIFIC ESCALAB Xi+, Waltham, MA, USA) type was as follows: monochromatic Al target (E *=* 1486.68 eV), Voltage: 14,795.40 V, Current: 0.0108 A, Vacuum: *p* < 10^−^^9^ mBar, Pass Energy: 100 eV (Survey), 20 eV (High-resolutions), Work FN: 5.04 eV. Transmission electron microscopy (TEM: JEM-2100Plus, Nippon Electron Corporation) model was elevated to 200 kV, and the ion pump readings were less than 2 × 10^−^^5^ Pa.

### 2.2. High-Temperature and High-Pressure Corrosion Experiments

#### 2.2.1. High-Temperature and High-Pressure Corrosion Conditions

The experimental conditions are shown in Table 4.

#### 2.2.2. High-Temperature and High-Pressure Corrosion Steps

Before the high-temperature and high-pressure corrosion test, the surface of the specimens was polished from 400^#^ to 1500^#^ SiC sandpaper step by step, to ensure that the surface roughness of the specimen was less than 1.6 μm. The polished specimens were put into acetone solution and cleaned with ultrasonic instrument for 3–5 min to remove the oil and impurities on the surface of the specimens, and then put into anhydrous ethanol and cleaned with ultrasonic instrument for 3 min. Then, the specimens were put into anhydrous ethanol for 3 min to achieve the purpose of dehydration, then blown dry with cold air, weighed, and the quality was accurate to 0.0001 g. The length, width, and height of the specimens were measured, and the size was accurate to 0.01 mm; finally, the specimens were saved in a dry vessel for use.

The specimens were stressed with the four-point bending fixture before being put into the simulated autoclave (350 ± 1 °C, and 35 ± 0.5 MPa, Dalian Kemao Experimental Equipment Company, Dalian, China); five parallel TC4 titanium alloy specimens, each in experimental condition, were submerged in the corrosive medium, and then quickly sealed. The autoclave was installed and deoxygenated with high purity N_2_ for 3 h to form an oxygen-free environment inside the autoclave. After the end of deaeration, the temperature started to increase; when the temperature reached the setting value, CO_2_ gas was passed into the autoclave. After test lasting for 7 d, the specimens were cleaned and weighed again.

### 2.3. Electrochemical Experiments

#### 2.3.1. Electrochemical Conditions

To investigate the effect of different stresses on the corrosion behavior of TC4 titanium alloy, the experimental conditions are shown in Table 5.

#### 2.3.2. Electrochemical Steps

The electrochemical test was carried out by an electrochemical workstation (Potentiostat P4000, Ametek Inc., Berwyn, IL, USA) with a standard three-electrode system; among them, TC4 titanium alloy was used as the working electrode, PTFE silver chloride electrode was used as the reference electrode, and Pt electrode was used as the auxiliary electrode. Before the test, the surface of the specimen was sandpapered to 2000^#^ to make the surface roughness meet the test requirements, and deoiled by acetone, dehydrated by anhydrous ethanol, and dried with cold air. The surface of the specimen and stress fixture were sealed with epoxy AB glue, leaving only a 10 mm × 10 mm working area of the specimen. N_2_ (99.99%) was used to remove oxygen for 1 h, then the corresponding gas was introduced and the temperature was increased to reach the set temperature. The working electrode was prepolarized at −1.2 V for 3 min to remove the oxide film formed on the electrode surface in the air; then, the working electrode was left in the solution for 0.5 h, and the open-circuit potential test was carried out after the system was stabilized for 1 h. After the open-circuit potential (OCP) was smooth, the electrochemical impedance spectroscopy (EIS) and polarization curve measurement started.

The EIS frequency-setting value was 10^−2^–10^5^ Hz, the measurement-signal amplitude was set to 10 mV sinewave, and the number of points throughout was 50.

The polarization-curve-test range was −1000~+1600 mV, and the scan-rate-setting value was 0.3333 mV/s.

The Mott–Schottky (M-S) curves were measured at 1000 Hz, the potential change interval was from −1.0 to −0.2 V, the AC amplitude was 5 mV, and the step potential was 5 mV. The polarization curves were analyzed and fitted using VersaStudio software(VersaStudio 2.1, Ametek Inc., Berwyn, IL, USA), and the impedance plots were analyzed and fitted using ZSimDemo impedance analysis software(ZSimDemo 3.60, Informer Technologies, Inc., Los Angeles, CA, USA). The polarization resistance was calculated as follows:(2)$$R_{P}=\frac{B_{a}\;\cdot \;B_{c}}{2.3\;\cdot \;\left(B_{a}+B_{c}\right)\;\cdot \;i_{\mathrm{corr}}}$$

## 3. Results and Discussion

### 3.1. Corrosion Behavior of Immersion Experiments

#### 3.1.1. Corrosion Morphology

Figure 2 and Figure 3 are the macroscopic morphology and microscopic morphology of TC4 titanium alloy specimens without cleaning formed under different stresses. From Figure 2, it can be seen that the surface of the specimen loaded with different stresses was accompanied by a small but varying amount of corrosion products; there were small pitting traces and the surface of the specimen was dull and lackluster. From Figure 3, it can be seen that a smaller number of corrosion products scattered on the surface of the specimens. Even if the stress applied to the specimen was 103% σ_s_, corrosion products were still relatively less, indicating that the TC4 titanium alloy was slightly soluble or suffered from slight corrosion under such conditions, and its corrosion resistance was better.

Figure 4 and Figure 5 are the microscopic morphology of TC4 titanium alloy specimens after cleaning under different stresses. From Figure 4, it can be seen that the surface of the specimens loaded with different loads did not present obvious traces of pits; the color of the specimens was nonmetallic luster and dull. It can be seen from Figure 5 that the pits on the surface of the specimens gradually increased with the increase in stress, and compared with other specimens in the elastic-deformation interval, the number of the pits on the surface of the specimen were more under stress loading of 20% σ_s_ and 103% σ_s_. When the applied stress was 103% σ_s_, the specimen was in the plastic deformation stage, and the pits on the surface of the specimen were deeper and wider.

To further observe the microscopic morphology of TC4 titanium alloy, TEM characterization of TC4 titanium alloy was carried out, as shown in Figure 6. It can be seen that the equiaxed grains of the specimen loaded with elastic stress had clear grain boundaries and a large number of dislocations. Meanwhile, the grains in some areas of the specimen loaded with plastic stress lost their original shape, the grain boundaries were blurred, and the dislocation lines were irregularly distributed with no obvious dislocation orientation, indicating that the specimen had a high density of dislocations and dislocation cells as well as an entanglement of dislocations. According to the von Mises criterion, more than five independent slip systems were required for the homogeneous plastic deformation of metals and alloys. In the case of densely arranged hexagonal (hcp) α phase, the prismatic and basal slip systems provided only four independent slip systems. Therefore, except for dislocation slip, the deformation of α phase required twinning to accommodate the plastic strain [30]. Therefore, the aggregation of these defects forced local breakage of the passivation film, and then the corrosion resistance of TC4 titanium alloy reduced.

#### 3.1.2. Corrosion Products

Figure 7 is the elemental composition the surface of TC4 titanium alloy loaded with 80% σ_s_ and 103% σ_s_ stresses at 200 °C, and its content is shown in Table 6. The main elements of the surface of the specimens loaded with elastic stress and plastic stress were O, Ti, Al, V, and a handful of NaCl crystals, i.e., Na and Cl elements, as derived from EDS analysis. Figure 8 shows X-ray diffraction of the surface of TC4 titanium alloy loaded with 80% σ_s_ and 103% σ_s_ stresses. It can be seen that the surface of the specimens was attached to α Al_2_O_3_ (Corundum, ICDD: 73-1512), and TiO_2_ (Anatase, ICDD: 71-1166), while it is possible that these oxides were only a thin layer attached to the surface of TC4 titanium alloy, but also presented a large amount of Ti (ICDD: 88-2321).

Figure 9 shows the XPS full spectra of TC4 titanium alloy specimens loaded with 80% σ_s_ and 103% σ_s_ stresses, respectively. It can be seen that the passivated film of TC4 titanium alloy constituted of Ti, V, Cl, Na, O, and C elements, etc. Among them, the highest intensities of the characteristic peaks were the O and C peak, indicating that the surface of TC4 titanium alloy had a high level of O and C; when the specimen of TC4 titanium alloy was polished by sandpaper before test, C detected in the spectrum was contaminated carbon, which was usually unavoidable for XPS analysis [31]. Ti was more reactive and facilitates the generation of an oxide film by O in the air after being placed in the air for a certain period of time, so that O was detected on the surface, and O was the oxidation reaction of TC4 titanium alloy with O_2_ to generate oxides. Since the content of Al and V in TC4 alloy was much lower than that of Ti, and the surface was covered by a self-generated oxide film, the number of corresponding oxides generated with O was low, so the characteristic peaks of Al and V in the film layer of TC4 titanium alloy were weaker. The results were also consistent with that of the EDS energy-spectrum analysis mentioned above. Furthermore, Ti, Al, and V on the surface of TC4 titanium alloy underwent different degrees of oxidation, but the oxide content of V on the surface was low and its characteristic peak intensity was not high, which was the reason why the characteristic peaks of oxide diffraction were not detected during the above XRD analysis.

Furthermore, the characteristic peaks of Ti, Al, and V are found in Figure 9. The results of the split-peak fitting of the three elements O, Ti, and Al are shown in Figure 10. O^2−^, OH^−^ and H_2_O are the forms of element O. The figure shows that the bond energy of O1s in the TC4 titanium alloy loaded with 80% σ_s_ was 532.3 eV, while the bond energy of O1s in the alloy loaded with 103% σ_s_, and the bond energy of O1s was 532.7 eV. The O element in the surface layer of TC4 titanium alloy should be presented mainly in the form of O^2−^, indicating the presence of TiO_2_ phase in the oxide film of TC4 titanium alloy [32]. The Ti2p 3/2 main peak and the Ti2p 1/2 companion peak position appeared at 458.1 eV and 465 ± 0.1 eV, respectively, combined with the O1s peak, further verifying the presence of the TiO_2_ “O-Ti-O O-Ti-O” bond of TiO_2_. In addition, a small amount of Al was detected on the surface of TC4 titanium alloy; the bond energy of Al2p in TC4 titanium alloy loaded with 80% σs was 74.3 eV, while the bond energy of Al2p in TC4 titanium alloy loaded with 103% σ_s_ was 74.7 eV. Compared with the bond-energy data, it can be determined that the Al in TC4 titanium alloy presented in the form of A1^3+^ corresponding to the Al_2_O_3_ bond, indicating the presence of A1_2_O_3_ phase in the obtained oxide film [33]. Compared to the loaded 80% σ_s_ TC4 titanium alloy, the amount of Al elements oxidized in TC4 titanium alloy loaded with 80% σ_s_ was higher, and more A1_2_O_3_ was generated, which was consistent with the XRD characterization results described above.

Therefore, during the loading stress of the TC4 titanium alloy, Al might migrate outward and react with O or Cl^−^. However, since the content of Al elements in the TC4 titanium alloy was significantly lower than the content of Ti elements, there was still a certain amount of Ti in the corresponding area; thus, there were two phases of Al_2_O_3_ and TiO_2_ presented in the oxide film.

### 3.2. Electrochemical Experimental Characteristics

#### 3.2.1. Polarization Curve

The polarization curves of TC4 titanium alloy loaded with different stresses are shown in Figure 11, and the fitted results are listed in Table 7. It can be seen that the self-corrosion potential E_corr_ of TC4 titanium alloy loaded with different stresses did not increase with the increase in loading stress. When the applied stress was in the elastic-deformation interval, E_corr_ of TC4 titanium alloy showed the trend of increasing first and then decreasing with the load of different stresses; however, E_corr_ of TC4 titanium alloy loaded with smaller stress (20% σ_s_) was relatively larger, while when the applied stress was in the plastic interval, E_corr_ of TC4 titanium alloy was largest.

E_corr_ reflected the tendency of the possibility of corrosion but did not indicate the rapidity of the corrosion rate [34]. In contrast, the corrosion rate of the electrode material was related to the corrosion-current density i_corr_, and the higher the corrosion-current density, the faster the corrosion rate [35]. Since the corrosion-current density presented a pattern of 40% σ_s_ < 60% σ_s_ < 80% σ_s_ < 20% σ_s_ < 103% σ_s_, among them, the corrosion-current density in the elastic-deformation interval with 20% σ_s_ condition was one order of magnitude higher than that in the other three conditions, and the loading stress in the plastic deformation interval (103% σ_s_) was one order of magnitude higher than that in the elastic-deformation interval (20% σ_s_). The polarization resistance R_p_ was opposite to the corrosion-current density i_corr_, indicating that the magnitude of the corrosion resistance law for TC4 titanium alloy loaded with different stress loads was as follows: 40% σ_s_ > 60% σ_s_ > 80% σ_s_ > 20% σ_s_ > 103% σ_s_.

#### 3.2.2. AC Impedance

Figure 12 shows the EIS results of TC4 titanium alloy loaded with different stresses. It can be seen that in the elastic-deformation stage, TC4 titanium alloy showed double capacitance-resistance arc characteristics, the high-frequency capacitive reactance arc reflected the electrochemical reaction process on the electrode surface, while the low-frequency capacitive reactance arc reflected the oxide film formed on the alloy surface when the alloy was dissolved. Furthermore, the larger the Nyquist impedance arc, the better the corrosion resistance of TC4 titanium alloy [36]. The radius of the low-frequency capacitance-resistance arc decreased with different loading stresses, indicating the decrease in corrosion resistance. In the plastic deformation stage, the radius of the low-frequency capacitance-resistance arc further decreased and Warburg impedance characteristics appeared. To visually obtain the corrosion resistance characteristics of TC4 titanium alloy loaded with different stresses, the impedance data were fitted by ZSimDemo analysis software, and the equivalent circuit diagram is shown in Figure 13, where R_s_ represents the solution resistance, C_c_ and R_c_ are the electrode-surface ion-adsorbed double electric-layer capacitance and electrode-surface ion-adsorbed bilayer resistance, respectively; C_dl_ and R_ct_ are the double electric-layer capacitance and charge-transfer resistance, respectively; and W is the Warburg diffusion resistance. The fitting results are shown in Table 8.

As can be seen from Table 8, in the elastic-deformation stage, the value of R_c_, R_ct_, C_dl_, and C_c_ increased first and then decreased with the increase in stress. Usually, elastic tensile stress decreased the pitting resistance of the alloy [37], indicating that the changes in the stability of alloy crystals might be the reason for the decrease in the pitting resistance of the alloy loaded with smaller elastic tensile stresses. Furthermore, the presence of point defects (vacancies, interstitial atoms, etc.) in alloy crystals increased the alloy crystal energy, and then increased the crystal instability. However, defects also led to an increase in the crystal entropy value, and the higher the entropy value, the more stable the crystal [38]. Therefore, the corrosion resistance of TC4 titanium alloy loaded with different stress loads showed a pattern of 40% σ_s_ > 60% σ_s_ > 80% σ_s_ > 20% σ_s_ > 103% σ_s_ due to the interaction of multiple factors mentioned above. This result was consistent with the polarization-curve results.

#### 3.2.3. M-S Curve

Figure 14 shows the M-S curves of TC4 titanium alloy loaded with different stress. As can be seen from the figure, the M-S curves of TC4 titanium alloy loaded with elastic stresses (20% σ_s_, 40% σ_s_, 60% σ_s_ and 80% σ_s_) were in the first quadrant, while the M-S curve of TC4 titanium alloy loaded with plastic stresses (103% σ_s_) was obviously in the second quadrant.

The semiconductor properties of metal passivation films can be described by Mott-Schottky theory [39,40].
(3)$$n-\mathrm{type}\;\mathrm{semiconductor}\;\mathrm{film}:\;\frac{1}{C_{SC}^{2}}=\frac{2}{\varepsilon \varepsilon_{0}eN_{D}}\left[E-E_{FB}-\frac{kT}{e}\right]$$
(4)$$p-\mathrm{type}\;\mathrm{semiconductor}\;\mathrm{film}:\;\frac{1}{C_{SC}^{2}}=\frac{-2}{\varepsilon \varepsilon_{0}eN_{A}}\left[E-E_{FB}-\frac{kT}{e}\right]$$
where, *C**_SC_*, space-charge-layer capacitance of the semiconductor film; *ε*_0_, vacuum capacitance, *ε*_0_ = 8.85 × 10^−12^ F·m^−1^;*ε*, the dielectric constant of the passivated film at room temperature, $\varepsilon_{{\mathrm{TiO}}_{2}}$ = 114;*N_D_*, the applied host concentration;*N_A_*, recipient concentration;*E*, potential of the reference electrode on the absolute scale; *E_FB_*, flat-band potential;*k*, Boltzmann constant, *k* = 1.38 × 10^−23^;*T*, thermodynamic temperature;*e*, electron charge, *e* = 1.602 × 10^−19^ C.


At room temperature, *kT*/*e* = 25 mV, so it can be neglected.

The capacitance value obtained from the test can be directly set to the space-charge-layer capacitance *C_SC_*, since other series capacitances such as the Hemholtz capacitance can be ignored, and the plot of *C_SC_*^−2^ against potential (*E*) is the Mott–Schottky plot. From the slope of the line, the type of semiconductor (n-type or p-type) of the passivation film can be deduced, and from the slope of the line and the intercept of the line on the potential axis, the applied concentration *N_D_* or the applied concentration *N_A_* can be found.

It can be seen that the passivation film of TC4 titanium alloy loaded with elastic stresses presented an n-type semiconductor characteristic with cation-selective permeability, which effectively prevented Cl^−^ from contacting with the matrix through the passivation film, showing the strong corrosion resistance of the passivation film from the corrosive medium. Under plastic-deformation conditions, the passivation film presented a p-type semiconductor with anion-selective permeability; thus, Ti, Al, and V in the matrix in the form of the corresponding anion could easily migrate to the surface of the passivation film.

Taking the linearity of the interval R2, the acceptor concentration *N_A_* and donor concentrations *N_D_* within the passivation film of TC4 titanium alloy loaded with different stresses were derived according to the Mott–Schottky relationship equation listed in Table 9. Analysis of the Mott–Schottky equation showed that when TC4 titanium alloy was loaded with stresses in the elastic deformation interval and *N*_D_ value of the passivation film under less and more stresses was larger while the loaded stresses were at intermediate values, the *N*_D_ value of the passivation film was smaller and *N*_D_ values were specifically arranged as 20% σ_s_ > 80% σ_s_ > 60% σ_s_ > 40% σ_s_. The larger the value of *N*_D_ and *N*_A_ in the passivation film, the faster the diffusion of ions and the greater the corrosion rate of the matrix [41]. Thus, the corrosion resistance of TC4 titanium alloy in this environment was arranged as 20% σ_s_ < 80% σ_s_ < 60% σ_s_ < 40% σ_s_. This is consistent with the above EIS and polarization-curve results.

In contrast, the slope of Mott–Schottky for TC4 titanium alloy loaded with 103% σ_s_ stress was significantly negative, indicating that the passivated film exhibited p-type semiconducting characteristics. The reason for the change in the semiconducting characteristics of the passivated film was closely related to its film components under different stress conditions. Usually, titanium oxides [42,43] and iron oxides [44] have n-type semiconductor characteristics, while aluminum oxides [45] and silicon oxides have p-type semiconductor characteristics [46]. According to the XPS analysis, the passivation film of TC4 titanium alloy loaded with 103% σ_s_ stress was mainly composed of two phases, Al_2_O_3_ and a small amount of TiO_2_, and its passivation-film semiconductor characteristics were influenced by the Al_2_O_3_ phase, thus exhibiting p-type semiconductor characteristics, which is in good agreement with the EDS and XRD results.

## 4. Corrosion Mechanisms

The formation processes of pits could be reliably described by adsorption and incorporation theories [47]. The surface passivation film of TC4 titanium alloy was mainly composed of TiO_2_ in air, while when TC4 titanium alloy was immersed in dielectric solution, Cl^−^ preferentially adhered on the surface of the surface-passivation film of TC4 titanium alloy, and entered the passivation film at the local discontinuity of the passivation film due to the small radius of Cl^−^; thus, the passivation film was tarnished, then an intense induction-ion conductivity was formed, and the passivation film at certain local points kept a higher current density. When the corrosion conditions became more and more severe, the current density on the surface of the specimen increased, and the high current density enabled the cations in the specimen to jump around disorderly and act flexibly, and when the electric field at the interface between the passivation film and the solution reached a certain critical value, and then pitting corrosion occurred. The coupling between them caused the formation of an aggressive environment because the dissolution of the Ti element in the matrix increased the activity of Ti^4+^, and the Cl^−^ ion migrated toward the defect location on the surface of the passivation film or/and TC4 titanium alloy. The hydrolysis reaction increased the local acidity so that the passivation film was eroded, and then the TC4 titanium alloy matrix was corroded. In addition, the repassivation rate at the bottom of the defect was less than that of TC4 titanium alloy; the rate of repassivation at the bottom of the defect was less than the dissolution rate of TC4 titanium alloy; and the equilibrium of “rupture-repair” could not be maintained, resulting in the formation of pitting nuclei [48,49]. As shown in Section 3.1, TC4 titanium alloy in a specific corrosive environment was characterized, and it was found that as the corrosive environment became more severe, the number of pits on the surface of the specimen gradually increased, and the size and depth of the pits gradually increased.

Under the elastic stress, TC4 titanium alloy deformed so that the passivation film was more likely to be ruptured, and the reaction activity of cations also increased, which promoted the corrosion processes. When TC4 titanium alloy was subjected to elastic-stress stretching, the absolute value of positive and negative external pressure increased, the chemical sites of the metal atoms excited by the deformation increased, and the activity of TC4 titanium alloy increased. Furthermore, the tensile stress also increased the grain clearance; there was a tendency to shift from the dense row to the non-dense row surface, providing energy for its corrosion behavior. TC4 titanium alloy underwent elastic deformation; the change in chemical sites led the TC4 titanium alloy equilibrium potential and electrode potential to shift negatively, which meant that the tendency of TC4 titanium alloy to be oxidized increased, and the tendency of metal autolysis also increased. As the electrode potential of the elastically deformed TC4 titanium alloy became negative, TC4 titanium alloy was corroded as the anode of the corrosion cell, and the electrode potential became negative, so the electric potential of the corrosion microcell increased and the circuit corrosion current increased; thus, the corrosion of TC4 titanium alloy in the elastic stress state was accelerated [50]. Roh and Macdonald [51] proposed that defects depleted at the passivation-film/medium-solution interface by migrating from the interface between the TC4 titanium alloy and passivation film to the interface between the passivation film and medium solution through certain reactions, such as the interaction of oxygen vacancies in the defects with water to produce a defect located at oxygen and hydrogen ions in the anionic lattice position, improving the corrosion resistance of TC4 titanium alloy. It is also possible that the dislocation shifting in TC4 titanium alloy made the interatomic gap be smaller, resulting in smaller vacancies and better corrosion resistance than that of TC4 titanium alloy loaded with 20% σ_s_. When the value of loading stress continuously increased, the passivation film on the surface of the TC4 titanium alloy ruptured while the angle of dislocation of the crystals shifted, the outcrop-per-unit area of the metal surface increased, and the area and probability of contact between the aggressive ions and the substrate increased greatly; thus, the corrosion resistance of TC4 titanium alloy decreased.

When the stress increased to the plastic interval, the stress in the plastic interval maybe caused defects and dislocations to migrate, causing a change of the cell, and then defects such as dislocation hair clusters and crystal surfaces slipped, which were favorable for the location of the pitting nucleation, further increasing the corrosion area and accelerating the corrosion processes [52]. When the loading stress was 103% σ_s_, on the one hand, the stability of the passivation film on TC4 titanium alloy reduced, and the film surface locally ruptured. On the other hand, the matrix was subject to plastic stress and the surface produced a large number of outcrops and even slipping steps, as shown in Figure 6. When coupling in the same location, corrosive ions gathered in the outcrops or slip steps, TC4 titanium alloy matrix contacted with the medium ions, and the TC4 titanium alloy/Cl^−^ solution interface was formed, at which plastic stress increased the anodic activity of TC4 titanium alloy in the solution containing Cl^−^ and promoted anodic dissolution. Meanwhile, repassivation of the film layer maybe also occurred, and the “passivation film rupture-repair “ processes also occurred locally in the passivated film. The cyclic processes reduced the corrosion resistance of TC4 titanium alloy when TC4 titanium alloy was always exposed to the environment of Cl^−^ medium solution [53,54,55].

When the specimen was in plastic deformation, the corrosion mechanism was schematically shown in Figure 15. When CO_2_ gas was introduced, ${CO}_{3}^{2-}$ and ${HCO}_{3}^{-}$ were added to the medium solution, and ${CO}_{3}^{2-}$ had strong interfacial bonding energy with TiO_2_, strong surface-adsorption ability, and stronger erosion [56]; and when the passivation film was broken, corrosive ions such as Cl^−^ and H^+^ competitively absorbed the matrix, and then the hydrolysis of Ti^4+^ or/and Al^3+^ was also strengthened with the increase in temperature, which was conducive to the autocatalytic effect of acidification; TC4 titanium alloy was susceptible to pitting corrosion in the end [57].

The corrosion processes on the oxide layer proceeded according to the following sequence of reactions:

The anode reactions:(5)$$\mathrm{Ti}\;\to {\;\mathrm{Ti}}^{4+}{+4e}^{-}$$
(6)$$\mathrm{Al}\to {\mathrm{Al}}^{3+}{+3e}^{-}$$
followed by:Ti^4+^ + 4H_2_O → Ti(OH)_4_ + 4H^+^(7)
Ti(OH)_4_ + 3Cl^−^ → TiCl_4_ + 4OH^−^(8)
Al^3+^ + 3H_2_O → Al(OH)_3_ + 3H^+^(9)
Al(OH)_3_ + 3Cl^−^ → AlCl_3_ + 3OH^−^(10)

The cathode reactions:(11)$$2{\mathrm{CO}}_{2}+2H_{2}O\to {\mathrm{HCO}}_{3}^{-}+{\mathrm{CO}}_{3}^{2-}+3H^{+}$$
(12)$${\mathrm{HCO}}_{3}^{-}\to {\mathrm{CO}}_{3}^{2-}+H^{+}$$
2H^+^ + 2e^−^ → H_2_↑(13)

When the radius of the pits gradually became larger and the depth increased, the concentration of H^+^ per unit area at the bottom of the pits was higher, and in order to maintain the electrical neutrality, Cl^−^ migrated to the tip of the pits, further promoting the dissolution of the alloy and the acidification of the solution and inhibiting the regeneration of the passivation film [58]. Furthermore, the processes might be accompanied by intergranular corrosion due to the different potential between α phase and β phase, and grains and grain boundaries. At the same time, the large accumulation of H^+^ generated stress concentrations, and cracks would subsequently generate.

## 5. Conclusions

(1)Corrosion resistance of TC4 titanium alloy loaded with different stresses was in the following order: 40% σ_s_ > 60% σ_s_ > 80% σ_s_ > 20% σ_s_ > 103% σ_s_, among them the corrosion rate of TC4 titanium alloy loaded with 20% σ_s_ was an order of magnitude higher than that loaded with other elastic stresses, and the corrosion rate of TC4 titanium alloy loaded with 103% σ_s_ was a further order of magnitude higher.(2)In the elastic-deformation stage, TC4 titanium alloys showed double capacitance arc-resistance characteristics, the radius of the low-frequency capacitance arc further decreased, and Warburg impedance characteristics appeared when TC4 titanium alloys was loaded 103% σ_s_, indicating the changed corrosion mechanism.(3)The semiconductor properties of TC4 titanium alloy from n-type to p-type as deformation from elastic to plastic due to the change in components of passivation film and *N*_D_ value in the passivation film increased, which reduced the protection of film layer to the matrix, resulting in the more serious pitting corrosion.

## Figures and Tables

**Figure 1 materials-15-04381-f001:**
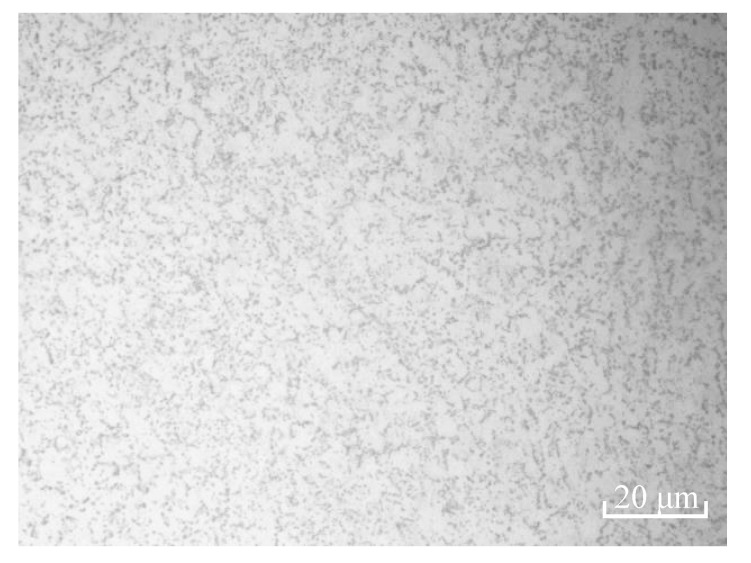
Microstructure of TC4 titanium alloy.

**Figure 2 materials-15-04381-f002:**
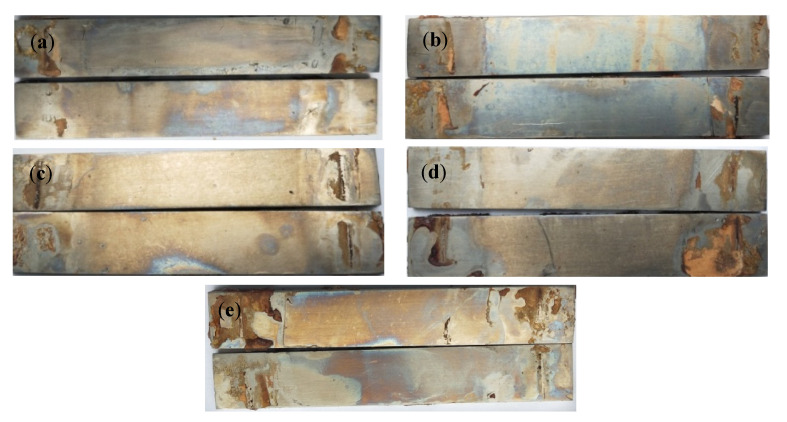
TC4 titanium alloy specimens under different stresses without cleaning the macroscopic corrosion morphology (**a**) 20% σs; (**b**) 40% σs; (**c**) 60% σs; (**d**) 80% σs; (**e**) 103% σs.

**Figure 3 materials-15-04381-f003:**
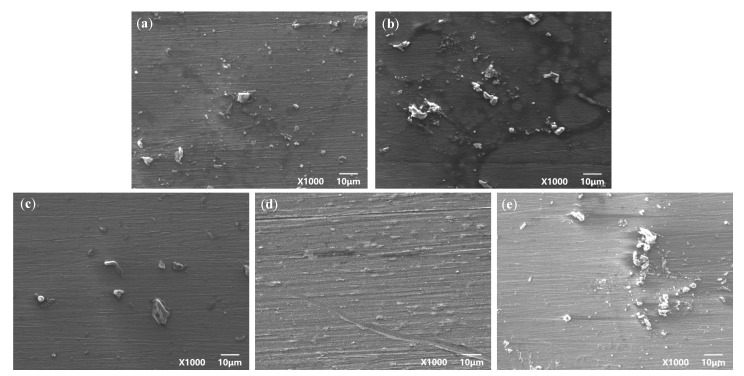
TC4 titanium alloy specimens under different stresses without cleaning the microcosmic corrosion morphology (**a**) 20% σs; (**b**) 40% σs; (**c**) 60% σs; (**d**) 80% σs; (**e**) 103% σs.

**Figure 4 materials-15-04381-f004:**
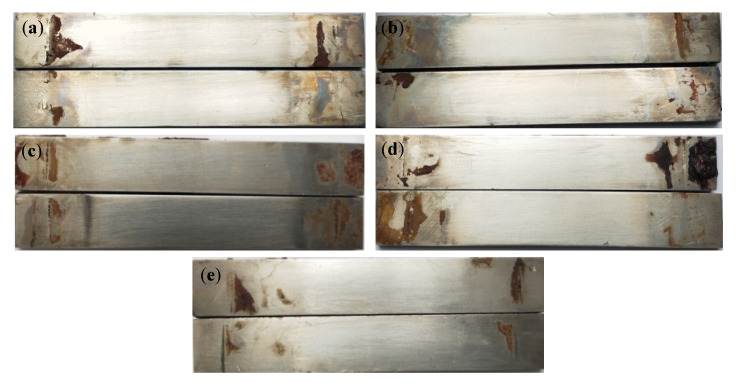
TC4 titanium alloy specimens in different stresses after cleaning the macroscopic corrosion morphology. (**a**) 20% σs; (**b**) 40% σs; (**c**) 60% σs; (**d**) 80% σs; (**e**) 103% σs.

**Figure 5 materials-15-04381-f005:**
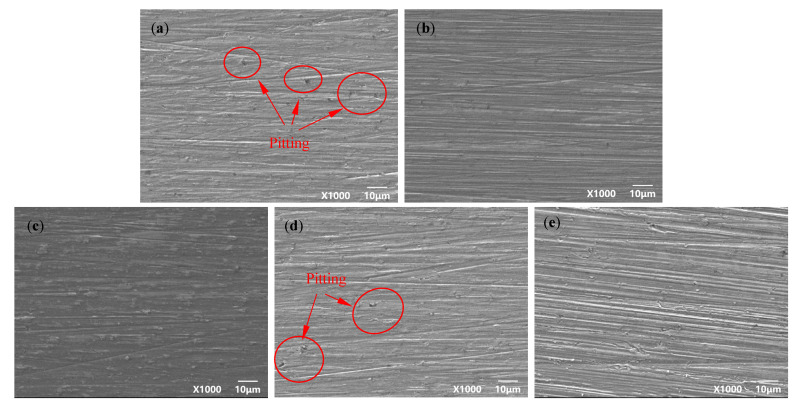
TC4 titanium alloy specimens in different stresses after cleaning the microscopic corrosion morphology (**a**) 20% σs; (**b**) 40% σs; (**c**) 60% σs; (**d**) 80% σs; (**e**) 103% σs.

**Figure 6 materials-15-04381-f006:**
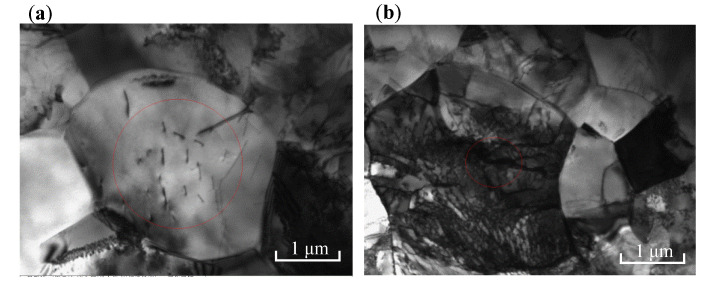
TEM morphology of TC4 titanium alloy (**a**) 80% σs; (**b**) 103% σs.

**Figure 7 materials-15-04381-f007:**
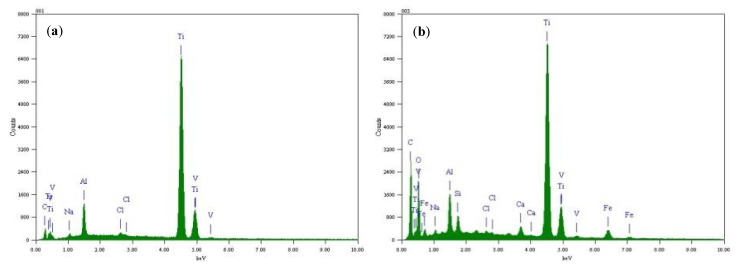
Location and results of EDS analysis of surface-corrosion products of TC4 titanium alloy loading different stresses (**a**) 80% σs and (**b**) 103% σs.

**Figure 8 materials-15-04381-f008:**
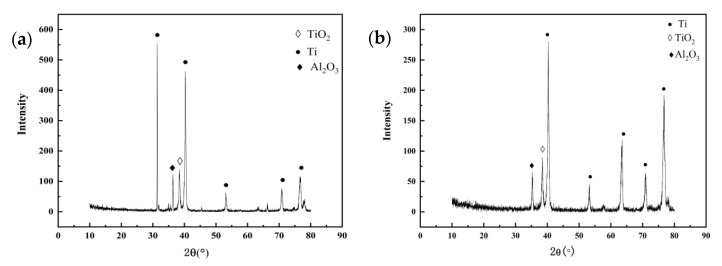
XRD results of surface-corrosion products of TC4 titanium alloy loaded with (**a**) 80% σs and (**b**) 103% σs stress.

**Figure 9 materials-15-04381-f009:**
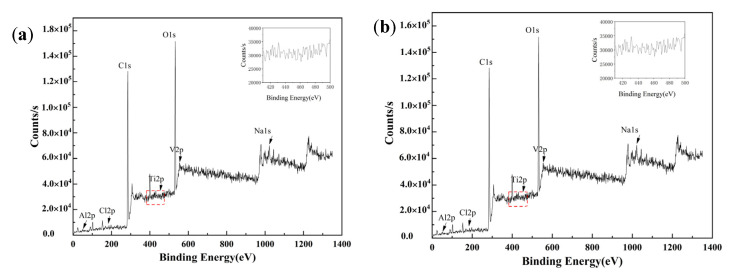
XPS full spectrum of TC4 titanium alloy loaded with (**a**) 80% σs and (**b**) 103% σs stress.

**Figure 10 materials-15-04381-f010:**
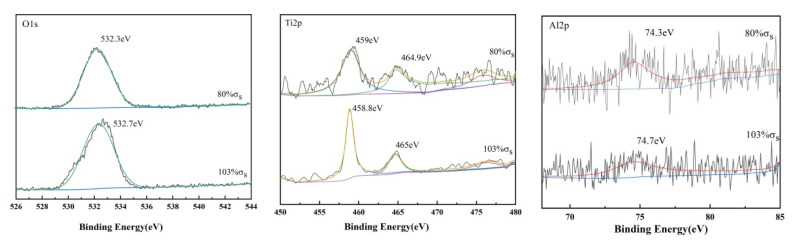
High-resolution XPS pattern of TC4 titanium alloy.

**Figure 11 materials-15-04381-f011:**
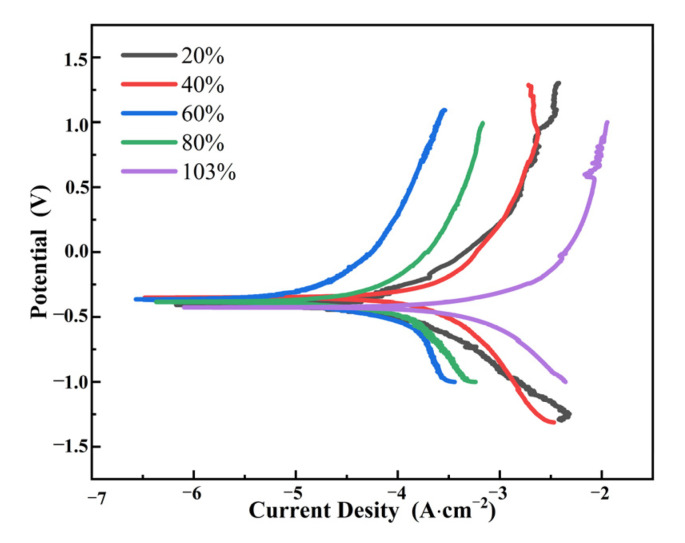
Polarization curves of TC4 titanium alloy with different loading stresses.

**Figure 12 materials-15-04381-f012:**
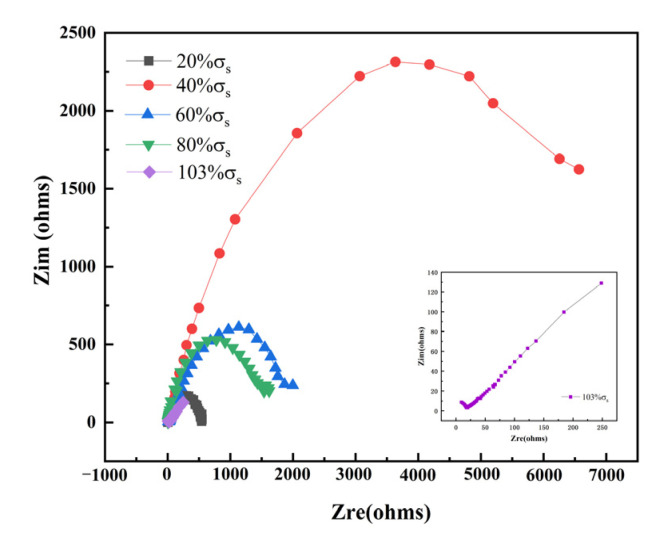
AC impedance spectrum of TC4 titanium alloy loaded with different stresses.

**Figure 13 materials-15-04381-f013:**
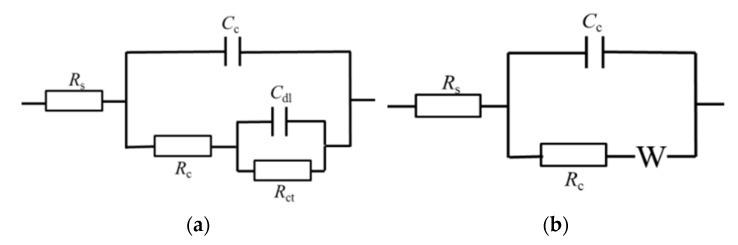
Equivalent circuit diagram. (**a**) Elastic deformation; (**b**) plastic deformation.

**Figure 14 materials-15-04381-f014:**
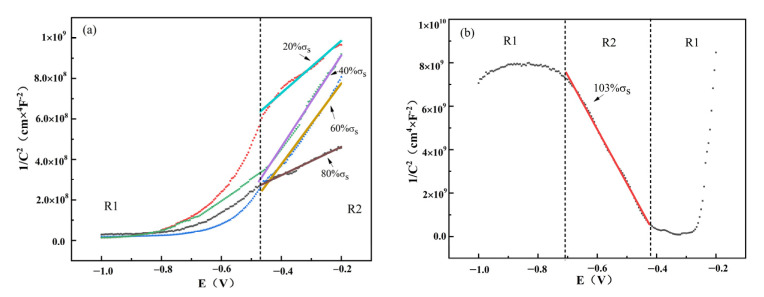
M-S curves of TC4 titanium alloy loaded with different stresses. (**a**) Elastic stress; (**b**) plastic stress.

**Figure 15 materials-15-04381-f015:**
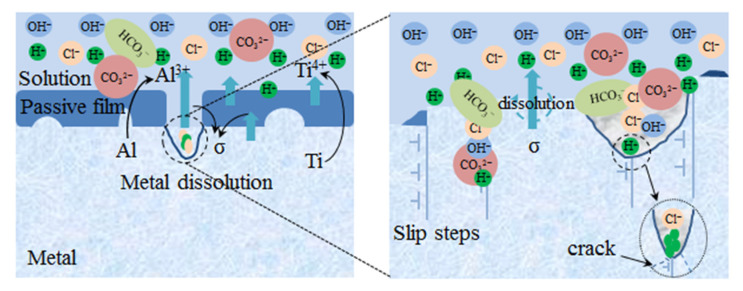
Plastic-deformation-zone corrosion-mechanism diagram.

**Table 1 materials-15-04381-t001:** Main chemical composition of TC4 titanium alloy (wt.%).

Element	Al	V	Fe	Si	C	O	N	H	Ti
Content	6.80	4.00	0.30	0.13	0.10	0.20	0.05	0.015	Balance

**Table 2 materials-15-04381-t002:** Parameters related to the mechanical properties of TC4 titanium alloy.

Sample	Plastic Elongation Strength/MPa	Tensile Strength/MPa	Elongation after Break/%	R_m_/MPa	R_p0.2_/MPa	R_p0.1_/MPa	A/%	Z/%	E/GPa
TC4	1130	1171	11.27	967	860	836	16.2	44.1	110

**Table 3 materials-15-04381-t003:** Specific values of applied stress.

Applied Load	20% σ_s_	40% σ_s_	60% σ_s_	80% σ_s_	103% σ_s_
Value/MPa	172	344	516	688	885.8
y/mm	0.323	0.646	0.969	1.292	1.663

**Table 4 materials-15-04381-t004:** Experimental parameters for corrosion evaluation under different stresses.

Conditions	Parameters				
Applied stress/MPa	20% σ_s_	40% σ_s_	60% σ_s_	80% σ_s_	103% σ_s_
Cl^−^ concentration/mg/L	25,000				
Total pressure/MPa	10				
CO_2_ partial pressure/MPa	4				
Temperature/°C	200				

**Table 5 materials-15-04381-t005:** Experimental conditions for corrosion evaluation of electrochemical test stress.

Conditions	Parameters				
Applied stress/MPa	20% σ_s_	40% σ_s_	60% σ_s_	80% σ_s_	103% σ_s_
Cl^−^ concentration/mg/L	25,000				
CO_2_ partial pressure/MPa	CO_2_ was continuously introduced during the experiment				
Temperature/°C	200				

**Table 6 materials-15-04381-t006:** Relevant parameters of element composition.

Element	80% σ_s_		103% σ_s_	
	Mass%	Atom%	Mass%	Atom%
C	9.44	26.87	8.70	22.65
O	3.80	8.11	8.33	16.18
Na	0.63	0.94	2.47	3.36
Al	4.49	5.69	6.37	9.06
Si	-	-	2.42	2.61
Cl	0.64	0.61	1.81	1.60
Ti	80.38	57.35	67.06	42.81
V	0.62	0.43	2.84	1.73
Total	100	100	100	100

**Table 7 materials-15-04381-t007:** Fitting results of polarization curves of TC4 titanium alloy with different loading stresses.

Corrosive Environments	Loaded Stress	E_corr_ (V)	i_corr_ (A·cm^−2^)	B_c_ (mV)	B_a_ (mV)	R_P_ (Ω·cm^−2^)
25,000 mg/L Cl^−^ 90 °C CO_2_	20% σ_s_	−0.413	1.168 × 10^−4^	27.548	89.865	7.85 × 10^4^
	40% σ_s_	−0.351	5.471 × 10^−5^	98.638	96.763	3.88 × 10^5^
	60% σ_s_	−0.358	5.122 × 10^−5^	34.666	36.221	1.50 × 10^5^
	80% σ_s_	−0.391	7.290 × 10^−5^	29.484	21.586	7.43 × 10^4^
	103% σ_s_	−0.426	6.759 × 10^−3^	21.364	36.486	8.67 × 10^5^

**Table 8 materials-15-04381-t008:** Fitting results of AC impedance spectra of TC4 titanium alloy loaded with different stresses.

Corrosive Environment	Applied Stress/(MPa)	R_s_/(Ω·cm^2^)	C_c_/(F·cm^−2^)	R_c_/(Ω·cm^2^)	C_dl_/(F·cm^−2^)	R_ct_/(Ω·cm^2^)	W/(Ω·cm^2^)
25,000 mg/L Cl^−^ CO_2_	20% σ_s_	3.189	3.476 × 10^−6^	65.34	1.344 × 10^−5^	427.1	/
	40% σ_s_	29.41	4.168 × 10^−5^	256.5	3.34 × 10^−4^	5691	/
	60% σ_s_	22.58	4.028 × 10^−5^	208.6	2.645 × 10^−4^	1468	/
	80% σ_s_	30.93	6.473 × 10^−6^	148.2	5.243 × 10^−5^	775.8	/
	103% σ_s_	20.13	6.306 × 10^−5^	19.36	/	/	1.483 × 10^−2^

**Table 9 materials-15-04381-t009:** M-S curves of TC4 titanium alloy loaded with different stresses.

Corrosive Environment	Loading Stress/MPa	*N*_D_/cm^−3^	*N*_A_/cm^−3^
25,000 mg/L Cl^−^ CO_2_ 90 °C	20% σ_s_	2.053 × 10^21^	1.763 × 10^21^
	40% σ_s_	9.051 × 10^20^	5.464 × 10^20^
	60% σ_s_	9.948 × 10^20^	9.565 × 10^20^
	80% σ_s_	1.203 × 10^21^	1.611 × 10^21^
	103% σ_s_	1.235 × 10^20^	4.965 × 10^19^

## Data Availability

The data presented in this study are available on request from the corresponding author.

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
