# Peer review of "Corrosion–Resistance Mechanism of TC4 Titanium Alloy under Different Stress-Loading Conditions"

_materials, 2022, doi:10.3390/ma15134381_

Round 1

Reviewer 1 Report

The paper is devoted to the problem of the stress-induced corrosion of Ti alloy TC4 (Ti 6Al 4V), which can be applied in the oil and gas industry. The paper presents interesting scientific results, but there are some significant comments to the paper text. The majority of comments are related to Section 2 Materials and Methods and Section 3 Results and Discussion.

1. Ref. [22], which discusses the effect of stresses on the passivating surface film for the austenite stainless steel 304 is not correct for Introduction. Similar remark concerns Refs. [17-19] in the Introduction. The present paper discusses the problem of corrosion resistance of the Ti alloys so the authors should provide the references to the works on the Ti alloys only.

22. In Table 1, the authors should indicate the exact chemical composition of the alloy under study, not the concentration intervals for the TC4 alloy.

33. The authors should describe the microstructure of the TC4 alloy under study in more details in the paper text paying attention to the sizes and shapes of the a- and b-phase particles. Is there any information about the alloy manufacturing technology? Who was the vendor of the alloy?

44. What is the quantity ss in Table 3? What quantity in Table 2 does this characteristic correspond to? The authors should define all the quantities, which are used in the paper text at the first mention.

55. With what uncertainty was the temperature maintained in the autoclave and how was it monitored? Are there any references to the works where the autoclave design was described in details?

66. What material were the devices made of for the stress-induced corrosion tests? The photos of the samples (Fig. 2) show clearly the area of contact corrosion – where the surface of the sample contacted the fixing point (support points). Why was there no isolation of the places of contact of the sample with the holder?

77. With what uncertainty was the measurement of the maximum deflection of the sample provided? How was this procedure carried out?

88. How long was the stress corrosion test carried out, the results of which are presented in paragraph 3.1? How long did the sample temperature control cycle take before the test?

99. How many samples were tested with each mechanical stress applied?

110.   What equipment was used for the electrochemical tests?

111.  Did the formation of hydrogen occur at the stage of preliminary polarization (-1.2 V, 3 min)?

112.  Why in Fig.2e do the surface colors for two samples of 103%ss differ so significantly from each other?

113.  In Fig. 5, the authors should present a photo of the sample, which was tested without the mechanical load applied. The presented photos of pitting raise doubts. The pit sizes are comparable to the ones of the scratches on the surface of the samples formed as a result of polishing.

114.  Is the location of the pitting related to any microstructural features of the TC 4 alloy?

115.  In which area on the surface of the sample were the studies carried out?

116.  In Section 2.1, the authors should describe the equipment used for the energy dispersion microanalysis.

117.  Please explain the presence of carbon in the EDS analysis results (see Table 6). If this is a consequence of using an oil pump, then the authors should normalize the EDS results. If this is a consequence of an incomplete surface cleaning (see XPS results analysis on page 8), then the results obtained are questionable – carbon can be embedded in the titanium oxide lattice at elevated temperature. This will affect the results obtained.

118.  It is not clear why NaCl crystals were present on the surface of the sample 103%ss, but weren’t present on the surface of the sample 80%ss.

119.  The XRD results presented in Figure 8 are highly questionable. It is almost impossible to detect the peak of titanium oxide by standard XRD measurements. In the case of deformed (curved) samples, the reliability of the results decreases even more. The authors should use GIXRD (Grazing Incident XRD) technique to asses the phase composition of the corrosion products on the samples’ surfaces. Please describe the equipment and the measurement modes used in the XRD analysis.

220.  In Fig. 8, the XRD peak from alumina is not visible. The presence of alumina on the Ti surface has not been proven (see the last sentence in the first paragraph of Subsubsection 3.1.2 on page 7).

221.  Why is there no NaCl peak in Fig. 8?

222.  The authors should describe, which phases of alumina and titania were detected.

223.  On page 8, it was stated that the tensile mechanical strain can affect the binding energy of atoms in the passivating surface film. At the same time, it was argued that an increase in the load will lead to a change in the binding energy. The scale of the change in the binding energy is 0.3-0.4 eV that is unreliable for the curved samples’ surfaces. The conclusion about the shift of the binding energy for Al made on the basis of the analysis of Fig.10 is particularly doubtful. The results of XPS need another interpretation.

224.  The conclusion that the Al atoms diffuse into the surface of the passivating film needs additional substantiation. Previously, it was shown that the XPS method allows detecting carbon contamination on the surfaces of the Ti samples (under the passivating film). Therefore, it seems more reasonable to assume that the formation of alumina occurred on the surface of the sample, not on the outer surface of the titania film.

225.  How was the experiment organized to obtain the polarization curves, which are shown in Fig. 11? How was the potential applied to the sample? What surface area was analyzed? Which compound was used to protect the surface? This procedure should be described in details in Section 2.

226.  How many times was the polarization curves i(E) shown in Fig.11 recorded? What was the reproducibility of the results? The potential values presented in Table 7 with the uncertainty of 0.001 are unreliable to the reviewer’s opinion.

227.   Please describe the method for determining the corrosion current density. Why does Table 7 show the data for the current, not for the current density?

228.  Are there any potential–time curves for the loaded samples?

229.   Why Table 7 and Table 9 indicate the temperature of 90 °C? In Table 4 and Table 5, which describe the experimental conditions the test temperature was indicated to be 200 oC. At what temperature was the experiment carried out actually?

330.  The model proposed in Section 4 is designed for pure Ti and does not take into account the two-phase structure of the TC4 alloy. The presence of the b-phase particles having a different potential can affect the electrochemical conditions of corrosion significantly. In the reviewer’s opinion, the authors should consider separately the behavior of the film on the surface of the a- and b-phases.

The reviewer does not have much experience with the AC impedance methodology, so he cannot assess the correctness of the results presented in Subsubsection 3.2.2.

Reviewer thinks, the paper needs major revision and re-review.

Author Response

Dear reviewer

    Thank you very much for your reviews!

  1. Ref. [22], which discusses the effect of stresses on the passivating surface film for the austenite stainless steel 304 is not correct for Introduction. Similar remark concerns Refs. [17-19] in the Introduction. The present paper discusses the problem of corrosion resistance of the Ti alloys so the authors should provide the references to the works on the Ti alloys only.

Response: Thank you for your suggestions. The text has been revised, and the responding references have been replaced. Please see Page 2 Introduction and Page 17 Reference section.

  1. In Table 1, the authors should indicate the exact chemical composition of the alloy under study, not the concentration intervals for the TC4 alloy.

Response: The exact chemical composition of TC4 ally used as the research object in the manuscript has been listed in Table 1, which was determined by OBLF QSN750 Photoelectric direct-reading spectrometer. Please see Table 1 in Page 3.

Table 1. Main chemical composition of TC4 titanium alloy (wt.%).

Element

Al

V

Fe

Si

C

O

N

H

Ti

Content

6.80

4.00

0.30

0.13

0.10

0.20

0.05

0.015

余量

  1. The authors should describe the microstructure of the TC4 alloy under study in more details in the paper text paying attention to the sizes and shapes of the α- and β-phase particles. Is there any information about the alloy manufacturing technology? Who was the vendor of the alloy?

Response: The description of the microstructure of TC4 alloy has been supplemented in the text, see subsection 2.1 Figure 1 in Page 1. The TC4 titanium alloy specimens used in this study were purchased directly from the manufacturer, and the supplier of the alloy is Shanghai Po Song Mechanical and Electrical Equipment Company.

20 um

Figure 1. Microstructure of TC4 titanium alloy

  1. What is the quantity σs in Table 3? What quantity in Table 2 does this characteristic correspond to? The authors should define all the quantities, which are used in the paper text at the first mention.

Response: The quantity σs in Table 3 is 860 MPa, σs stands for yield strength and corresponds to Rp0.2 in Table 2, which has been supplemented in the text, see page 3.

Table 3. Specific values of applied stress.

Applied load

20%σs

40%σs

60%σs

80%σs

103%σs

Value/MPa

172

344

516

688

885.8

y/mm

0.323

0.646

0.969

1.292

1.663

  1. With what uncertainty was the temperature maintained in the autoclave and how was it monitored? Are there any references to the works where the autoclave design was described in details?

Response: The temperature of the autoclave maintained in this study was 200±1 °C, which was monitored by the temperature monitor; the autoclave used in this experiment was manufactured in Dalian Kemao Experimental Equipment Company, its design temperature is 350±1 oC and design pressure is 35 MPa. The related information was added in the text, Please see Page 4.

  1. What material were the devices made of for the stress-induced corrosion tests? The photos of the samples (Fig. 2) show clearly the area of contact corrosion – where the surface of the sample contacted the fixing point (support points). Why was there no isolation of the places of contact of the sample with the holder?

Response: The fixture used to load the specimen with stress was a four-point bending stress fixture, as shown in the following figure, which was made of C276. There were also fixing points between the sample and the holder, which isolate the sample from the holder.

Where, t: thickness of specimen, mm

y: maximum deflection, mm

H: distance between external support points, mm

A: distance between the internal and external support points, mm

  1. With what uncertainty was the measurement of the maximum deflection of the sample provided? How was this procedure carried out?

Response: The measured value of sample deflection is y ± 0.001. The values of the deflection have been added to the text and are shown in Table 3. The screw of the four-point bending stress fixture rotates to apply force to the specimen, and the middle part of the specimen is raised, thus driving the rod that the specimen touches to move downward. The other end of this rod is connected to the display, and the deflection can be read by the display. As shown in following figure.

  1. How long was the stress corrosion test carried out, the results of which are presented in paragraph 3.1? How long did the sample temperature control cycle take before the test?

Response: Thanks for your suggestions. High-temperature and high-pressure corrosion experiments were conducted for a period of 7 days at the setting temperature and pressure, and the heating time of the autoclave to 200 oC before the test was generally around 4 hours.

  1. How many samples were tested with each mechanical stress applied?

Response: 5 samples for each mechanical stress were applied in each test.

  1. What equipment was used for the electrochemical tests?

Response: The entire electrochemical test was performed with the Potentiostat P4000 electrochemical workstation. As shown in following figure.

  1. Did the formation of hydrogen occur at the stage of preliminary polarization (-1.2 V, 3 min)?

Response: In the pre-experimental stage, the oxide film formed on the electrode surface in the air was dissolved and removed at lower voltage, hydrogen gas was not produced in this stage.

  1. Why in Fig. 2e do the surface colors for two samples of 103%σs differ so significantly from each other?

Response: Because there is a color difference between the two specimens when the light acts, there are five parallel specimens in this experiment, and the pictures in the manuscript have been replaced, please see Fig. 2e.

  1. In Fig. 5, the authors should present a photo of the sample, which was tested without the mechanical load applied. The presented photos of pitting raise doubts. The pit sizes are comparable to the ones of the scratches on the surface of the samples formed as a result of polishing.

Response: Firstly, we know that TC4 titanium alloy has excellent corrosion resistance; secondly, we conducted 3D characterization (super depth of field) of the specimen without loading stress and found that there are only tiny pitting pits on the surface of TC4 titanium alloy, but the depth of pitting pits did not reach the standard related to pitting depth and the depth of pits was failed to be characterized, as shown in the following figure. Furthermore, although the surface of the specimen loaded with stress has changed, there was no doubt about the pitting pits on its surface.

  1. Is the location of the pitting related to any microstructural features of the TC 4 alloy?

Response: The location of pitting may be related to the microstructure of TC4 titanium alloy. By reviewing the literature, J.R. Chen et al. tested in a mixed solution of 0.5 M H2SO4 and 1 M HCl and observed selective corrosion of the α phase at the α and β phase boundaries at open circuit potential. Also the higher dissolution rate of α-phase than β-phase was attributed to the fact that the passivation film formed on β-phase was more stable than that formed on α-phase. However, many other studies have indicated that the influence of variables such as α-phase and β-phase content and phase size on the corrosion resistance of titanium alloys is not clearly established, and there is still room and opportunity to further improve the corrosion performance of titanium alloys by effectively regulating the phase content.

Jhen-RongChen, Wen-TaTsai. In situ corrosion monitoring of Ti-6Al-4V alloy in H2SO4/HCl mixed solution using electrochemical AFM. Electrochimica Acta, 2011, 56(4): 1746-1751. https://doi.org/10.1016/j.electacta.2010.10.024

  1. In which area on the surface of the sample were the studies carried out?

Response: The study area on the surface of the specimen (The middle position between two inner fulcrums) was shown in the following Figure.

Study

area

  1. In Section 2.1, the authors should describe the equipment used for the energy dispersion microanalysis.

Response: Thanks for your suggestions. It has been supplemented in section 2.1 of the text. Please see Page 3.

  1. Please explain the presence of carbon in the EDS analysis results (see Table 6). If this is a consequence of using an oil pump, then the authors should normalize the EDS results. If this is a consequence of an incomplete surface cleaning (see XPS results analysis on page 8), then the results obtained are questionable – carbon can be embedded in the titanium oxide lattice at elevated temperature. This will affect the results obtained.

Response: The EDS results have been normalized in the text (as listed in Table 6 in Page 8). The presence of C in the EDS results analysis is due to the presence of rubber for sealing the rest of the sample, but the rubber has not yet decomposed and is a compound, which did not present in a molecule or an atomic state.

  1. It is not clear why NaCl crystals were present on the surface of the sample 103%σs, but weren’t present on the surface of the sample 80%σs.

Response: Because the selected area was randomized and the experiment had five parallel samples, the authors have replaced the figures and tables in the manuscript. Please see Figure7 and Table 6 in Page 8.

  1. The XRD results presented in Figure 8 are highly questionable. It is almost impossible to detect the peak of titanium oxide by standard XRD measurements. In the case of deformed (curved) samples, the reliability of the results decreases even more. The authors should use GIXRD (Grazing Incident XRD) technique to asset the phase composition of the corrosion products on the samples’ surfaces. Please describe the equipment and the measurement modes used in the XRD analysis.

Response: The equipment used for the analysis of XRD was an X-ray diffractometer (X'PertPro) with the model Shimadzu XRD-6000, with a continuous scanning mode, a scanning range of 5°-144° (2θ) and an X-ray tube power of 2.7 KW. In addition, the paper (Maria LuisaGrilli, MehmetYilmaz, SakirAydogan, et al. Room temperature deposition of XRD-amorphous TiO2 thin films: Investigation of device performance as a function of temperature, Ceramics International, 2018, 44(10): 11582-11590. https://doi.org/10.1016/j.ceramint.2018.03.222) showed the similar results of TiO2 composition.

  1. In Fig. 8, the XRD peak from alumina is not visible. The presence of alumina on the Ti surface has not been proven (see the last sentence in the first paragraph of Subsubsection 3.1.2 on page 7).

Response: The authors rechecked the XRD analysis results and detected a peak of alumina on the surface of the specimen, and the phase of Al2O3 is Corundum, which belongs to α-Al2O3. Please see Figure 8 in Page 9.

  1. Why is there no NaCl peak in Fig. 8?

Response: Firstly, there were fewer NaCl crystals deposited on the surface of the specimens due to the samples after test was not completely cleaned, and secondly NaCl was not a focus in this study.

  1. The authors should describe, which phases of alumina and titania were detected.

Response: The phase of Al2O3 detected in XRD is Corundum, which belongs to α- Al2O3. The phase of TiO2 is Anatase, and Titanium was additionally detected.

  1. On page 8, it was stated that the tensile mechanical strain can affect the binding energy of atoms in the passivating surface film. At the same time, it was argued that an increase in the load will lead to a change in the binding energy. The scale of the change in the binding energy is 0.3-0.4 eV that is unreliable for the curved samples’ surfaces. The conclusion about the shift of the binding energy for Al made on the basis of the analysis of Fig.10 is particularly doubtful. The results of XPS need another interpretation.

Response: The loading stress causes a change in the binding energy, but the range of change is not big. In addition, the amount of Al2O3 on the surface of the specimen is too small and the fitting results have also some errors. The XPS characterization in this study determined by the binding energy of the Al element is in the form of A13+, corresponding to the Al2O3 bond.

  1. The conclusion that the Al atoms diffuse into the surface of the passivating film needs additional substantiation. Previously, it was shown that the XPS method allows detecting carbon contamination on the surfaces of the Ti samples (under the passivating film). Therefore, it seems more reasonable to assume that the formation of alumina occurred on the surface of the sample, not on the outer surface of the titania film.

Response: First of all, TiO2 film layer on TC4 matrix is very thin, the area of Al2O3 formation is possible on the inner or outer surface of TiO2, and the area of Al2O3 migration is also random; so the Al content migrated outward.

  1. How was the experiment organized to obtain the polarization curves, which are shown in Fig. 11? How was the potential applied to the sample? What surface area was analyzed? Which compound was used to protect the surface? This procedure should be described in details in Section 2.

Response: In this study, stress was applied to the specimen by a four-point bending stress fixture. For electrochemical experiments, only the most central part of the specimen surface concentrated by the stress, i.e., a surface area of 10 mm × 10 mm, was studied. Epoxy resin adhesive was used to seal the fixture and other parts of the specimen to achieve the protective effect. So this content has been added in section 2, please see Page 4.

  1. How many times was the polarization curves i(E) shown in Fig.11 recorded? What was the reproducibility of the results? The potential values presented in Table 7 with the uncertainty of 0.001 are unreliable to the reviewer’s opinion.

Response: Each set of experiments was performed three times to eliminate errors in the experiments. The uncertainty values of the results were all varied within the range of 0.002.

  1. Please describe the method for determining the corrosion current density. Why does Table 7 show the data for the current, not for the current density?

Response: According to the steps described in subsection 2.3.2, the polarization curve was measured and fitted by the software that comes with the electrochemical workstation to obtain the self-corrosion potential and corrosion current density. Since the corrosion current density was the ratio of the corrosion current to the working surface area, and the working surface area of the electrode in this paper was 1 cm2, so the corrosion current was shown in Table 7. Now the corrosion current has been adjusted to the corrosion current density, see Table 7 in Page 11.

  1. Are there any potential–time curves for the loaded samples?

Response: The state of the specimen subjected to stress load in this study belongs to the constant load tensile state, focusing on the corrosion behavior of the specimen under the constant load state, so the rate of loading stress to the specimen is very fast, and potential–time curves test for the loaded samples was not carried out.

  1. Why Table 7 and Table 9 indicate the temperature of 90 °C? In Table 4 and Table 5, which describe the experimental conditions the test temperature was indicated to be 200 oC. At what temperature was the experiment carried out actually?

Response: This study was completed by two parts of tests: high-temperature and high-pressure immersion test and electrochemical test. The temperature of the high-temperature and high-pressure immersion experiment was 200 °C, and the temperature of the electrochemical test was 90 °C considering the stability of reference electrode.

  1. The model proposed in Section 4 is designed for pure Ti and does not take into account the two-phase structure of the TC4 alloy. The presence of the β-phase particles having a different potential can affect the electrochemical conditions of corrosion significantly. In the reviewer’s opinion, the authors should consider separately the behavior of the film on the surface of the α- and β-phases.

Response: As your said, there is the influence of phase fraction on the materials corrosion due to their different potential, which can be obtained by the SKPFM analysis (Liyang Zhu, Jiajia Wu, Dun Zhang, et al. Influence of the α fraction on 2205 duplex stainless steel corrosion affected by Pseudomonas aeruginosa. Corrosion Science 2021, 193: 109877) https://doi.org/10.1016/j.corsci.2021.109877. It is a great pity that relevant tests have not been done. And corrosion resistance of TC4 alloy is better than that of 2205 DSS, the influence of phase fraction on corrosion behavior of TC4 alloy may be small.

Reviewer 2 Report

The work is a well done basic experimental work, the paper is suitable for publication after some minor revisions.

In the following my suggestions to the Authors.

 As a general comment, Authors should pay more attention to writing (both typo and clarity of presentation) to enhance the work itself.

 Introduction

Line 37-38: too much “development” in two lines.

Line 44: after CO2 add the specification “content” or “concentration”.

Line 59: SCC is an acronym, each acronym must be specified the first time it is used.

 Materials

All the material used and equipments (TC4 alloy specimens, etc) need to be better detailed, as provider, company, measurements, etc.

Results

Figure 7: in the figure caption add the specification b).

Line 227: about the high carbon signal in the XPS spectrum, could it be related with the original C content in the alloys and to the CO2 fluxed in the autoclave under pressure?

Figure 12: the unit  is missing on the x axis.

Impedance measurements: Did the Authors try to fit the 103% sample with the circuit a), before change the equivalent circuit?

Author Response

Dear reviewer

    Thank you very much for your reviews!

Introduction

  1. Line 37-38: too much “development” in two lines.

Response: Thanks for your suggestions. The presentation has been revised in this manuscript.

  1. Line 44: after CO2 add the specification “content” or “concentration”.

Response: According to your advices, the specification “concentration” has been added after CO2.

  1. Line 59: SCC is an acronym, each acronym must be specified the first time it is used.

Response: According to your advices, SCC (Stress corrosion crack), including each acronym, has been specified the first time it is used.

Materials

  1. All the material used and equipments (TC4 alloy specimens, etc) need to be better detailed, as provider, company, measurements, etc.

Response: According to your advices, the manufacturer of the TC4 titanium alloy specimens in this study is Shanghai Po Song Mechanical Electrical Equipment Company. The model of SEM is JSM-6390A, the manufacturer is Japan Electronics Corporation, and EDS is an additional module used for composition analysis. The model of XRD is Shimadzu XRD-6000, the scanning range is 5°-144° (2θ), and the power of X-ray tube is 2.7KW. XPS model is Thermo SCIENTIFIC ESCALAB Xi+, type: monochromatic Al target (E=1486.68 eV), Voltage: 14795.40 V, Current: 0.0108 A, Vacuum: P<10-9 mBar, Pass Energy: 100 eV (Survey), 20 eV (High-resolutions), Work FN: 5.04 eV. The TEM model number is JEM-2100Plus and the manufacturer is Nippon Electron Corporation; the usage mode is elevated to 200 kV and the ion pump readings are less than 2X10-5 Pa. Added to the article. All of them have been added and better detaild. Please see Page 3.

Results

  1. Figure 7: in the figure caption add the specification b).

Response: According to your advices, the specification (b) has been added in the figure caption. Please see Figure 7.

  1. Line 227: about the high carbon signal in the XPS spectrum, could it be related with the original C content in the alloys and to the CO2 fluxed in the autoclave under pressure?

Response: The high carbon signal in the XPS spectra in this study mainly originated from trace amounts of epoxy resin adhesive on the surface of the specimens to seal the fixture, C content in the alloys is small, only 0.10wt.%, and the CO2 fluxed in the autoclave under pressure would form CO32-, the content of carbonate deposits is also less.

  1. Figure 12: the unit is missing on the x axis. Impedance measurements: Did the Authors try to fit the 103% sample with the circuit a), before change the equivalent circuit?

Response: I am very sorry for the errors, the unit on the x axis has been added. And the impedance spectrum loaded with 103% σs stress has Warburg resistance characteristics, so the equivalent circuit has been revised.

Reviewer 3 Report

Dear authors, I consider that your manuscript needs major revision. Please see the remarks presented in the attached review document.

Author Response

Dear Reviewer

    Thank you very much for your review!

  1. In the introduction, it is necessary to expand the general problem of the effect of aggressive chloride ions on metals, alloys and their protective coatings, which suffer from corrosion failure. For example: https://doi.org/10.1016/j.jallcom.2021.159309, https://doi.org/10.1016/j.corsci.2011.12.001.

Response: Thanks for your reviews and suggestions, the effect of aggressive chloride ions on metals was added. Please see Page 2.

  1. In paragraph 2.1, add SEM, TEM, EDS, XRD, XPS. Specify the manufacturers of the materials and equipment used, as well as the modes used in the study of materials. For example: XRD scan angles, scan speed, geometric type and etc.

Response: According to your advices, the manufacturer of the TC4 titanium alloy specimens in this study is Shanghai Po Song Mechanical Electrical Equipment Company. The model of SEM is JSM-6390A, the manufacturer is Japan Electronics Corporation, and EDS is an additional module used for composition analysis. The model of XRD is Shimadzu XRD-6000, the scanning range is 5°-144° (2θ), and the power of X-ray tube is 2.7KW. XPS model is Thermo SCIENTIFIC ESCALAB Xi+, type: monochromatic Al target (E=1486.68 eV), Voltage: 14795.40 V, Current: 0.0108 A, Vacuum: P<10-9 mBar, Pass Energy: 100 eV ( Survey), 20 eV (High-resolutions), Work FN: 5.04 eV. The TEM model number is JEM-2100Plus and the manufacturer is Nippon Electron Corporation; the usage mode is elevated to 200 kV and the ion pump readings are less than 2X10-5 Pa. all of them have been added in paragraph 2.1. Please see Page 3.

  1. Line 123 Specify the exact type of reference electrode used.

Response: Thanks for your suggestions. The reference electrode used in this study was a PTFE Silver chloride electrode. Please see Page 4.

  1. Figure 8 - Expand the description of the phase composition. Give the ICDD numbers for each of the found phases.

Response: According to your advice, the phase of Al2O3 detected in XRD is Corundum, which belongs to α- Al2O3 and its PDF card number is 73-1512. The phase of TiO2 is Anatase, its PDF card number is 71-1166. And the phase of Ti is Titanium, its PDF card number is 88-2321. All of them have been added in the manuscript. Please see Page 8.

  1. Paragraph 3.1.2 it is not very clear how different mechanical loads affect the change in bond energy by the XPS method. Perhaps you have identified a different oxide composition of corrosion products depending on the load? Please present the results in a clearer way.

Response: The present study is to determine the valence forms of the elements by comparing the bond energies of the product elements under different conditions, and thus the corresponding oxides were determined.

  1. Paragraph 3.2.1 - It's not very clear why you have jumps of Ecorr and icorr. It is necessary to explain the observed results, rather than a superficial analysis, that the current density is a quantitative characteristic of the corrosion rate. How do mechanical loads affect corrosion performance and why?

Response: Thanks for your suggestions. As the corrosion current density is the ratio of corrosion current to the working surface area and the working surface area of the working electrode in this paper is 1 cm2, so the corrosion current density was showed in Table 7. Now the corrosion current has been adjusted to the corrosion current density, listed in Table 7.

When TC4 titanium alloy is subjected to elastic stress stretching, the absolute value of positive and negative external pressure increases, the chemical sites of the metal atoms excited by the deformation increases, the activity of TC4 titanium alloy increases, while the tensile stress also increases the grain spacing, there is a tendency to shift from the dense row to the non-dense row surface, providing energy for its corrosion behavior. TC4 titanium alloy after elastic deformation, by the thermal potential, chemical site changes lead to TC4 titanium alloy equilibrium potential and electrode potential shift in the negative direction, which means that the tendency of TC4 titanium alloy to be oxidized increases, and the tendency of metal autolysis also increases. Becasue the elastic deformation of TC4 titanium alloy electrode potential becomes negative, TC4 titanium alloy, as the anode of the corrosion cell, will be corroded, and the electrode potential becomes negative, so that the electric potential of the corrosion microcell increases, the circuit corrosion current increases, thus the corrosion of TC4 titanium alloy in the elastic stress state is accelerated.

  1. Paragraph 3.2.2 - The processing of the received data was not carried out correctly. Table 8 incorrectly indicates the units of measurement of the Warburg element and does not show the accuracy of χ2 data approximation. ‘’Cdl and Rct are the bilayer capacitance and charge transfer resistance’’ are not correct definitions. Cdl and Rct are the double electric layer capacitance and charge transfer resistance.

Response: Thanks for your suggestions, the presentation have been revised. Please see subsection 3.2.2 and Table 8 for details.

  1. Looking closely at the data in Table 8 and Graph 12 for the 40%σs sample, the table value for charge transfer resistance is 1569 Ohm cm2, while in Figure 12 this value is over 7000 Ohm cm2.

Response: The data has been checked revised. Please see Table 8.

  1. To be able to compare the results obtained by the EIS method and polarization curves, it is necessary to calculate the polarization resistance Rp and give a formula for calculation.

Response: The data for polarization resistance (Table 7 in Page 11) and the calculation formula (2) have been added in the manuscript in Page 5.

  1. Paragraph 4 - The mechanism of pitting corrosion has been described previously. Please add links to the authors who have studied these interactions. For example: https://doi.org/10.1016/j.ceramint.2021.12.318

Response: Thanks for your suggestions, the mechanism of pitting corrosion was revised according to the corresponding references. Please see Pages 14-16.

  1. It is also necessary to add possible chemical and electrochemical reactions to the pitting corrosion scheme.

Response: According to your advices, the possible chemical and electrochemical reactions to the pitting corrosion scheme have been added. Please see Pages 15-16.

  1. Usually, in the process of pitting corrosion on the polarization curves in the semi-logarithmic coordinates, there is a break in the anodic curve, which is characterized by pitting. You don't see this break. It feels like the process is accompanied by intergranular corrosion, and pitting corrosion is not observed here.

Response: As you said, the break in the anodic curve is not obvious, the pitting process may be accompanied by intergranular corrosion due to the different potential between α-phase and β-phase, and grains and grain boundaries.

Reviewer 4 Report

Dear Authors,

Some adjustments need to be made.

1. In the introduction, it is necessary to expand the general problem of the effect of aggressive chloride ions on metals, alloys and their protective coatings, which suffer from corrosion failure. For example: https://doi.org/10.1016/j.jallcom.2021.159309, https://doi.org/10.1016/j.corsci.2011.12.001.

2. In paragraph 2.1, add SEM, TEM, EDS, XRD, XPS. Specify the manufacturers of the materials and equipment used, as well as the modes used in the study of materials. For example: XRD scan angles, scan speed, geometric type and etc.

3. Line 123 Specify the exact type of reference electrode used.

4. Figure 8 - Expand the description of the phase composition. Give the ICDD numbers for each of the found phases.

5. Paragraph 3.1.2 it is not very clear how different mechanical loads affect the change in bond energy by the XPS method. Perhaps you have identified a different oxide composition of corrosion products depending on the load? Please present the results in a clearer way.

6. Paragraph 3.2.1 - It's not very clear why you have jumps of Ecorr and icorr. It is necessary to explain the observed results, rather than a superficial analysis, that the current density is a quantitative characteristic of the corrosion rate. How do mechanical loads affect corrosion performance and why?

7. Paragraph 3.2.2 - The processing of the received data was not carried out correctly. Table 8 incorrectly indicates the units of measurement of the Warburg element and does not show the accuracy of χ2 data approximation. ‘’Cdl and Rct are the bilayer capacitance and charge transfer resistance’’ are not correct definitions. Cdl and Rct are the double electric layer capacitance and charge transfer resistance.

Looking closely at the data in Table 8 and Graph 12 for the 40%σs sample, the table value for charge transfer resistance is 1569 Ohm cm2, while in Figure 12 this value is over 7000 Ohm cm2.

To be able to compare the results obtained by the EIS method and polarization curves, it is necessary to calculate the polarization resistance Rp and give a formula for calculation.

8. Paragraph 4 - The mechanism of pitting corrosion has been described previously. Please add links to the authors who have studied these interactions. For example: https://doi.org/10.1016/j.ceramint.2021.12.318

It is also necessary to add possible chemical and electrochemical reactions to the pitting corrosion scheme.

Usually, in the process of pitting corrosion on the polarization curves in the semi-logarithmic coordinates, there is a break in the anodic curve, which is characterized by pitting. You don't see this break. It feels like the process is accompanied by intergranular corrosion, and pitting corrosion is not observed here.

Author Response

Dear reviewer

   Thank you very much for your review!!!

  1. In the introduction, it is necessary to expand the general problem of the effect of aggressive chloride ions on metals, alloys and their protective coatings, which suffer from corrosion failure. For example: https://doi.org/10.1016/j.jallcom.2021.159309, https://doi.org/10.1016/j.corsci.2011.12.001.

Response: Thanks for your reviews and suggestions, the effect of aggressive chloride ions on metals was added. Please see Page 2.

  1. In paragraph 2.1, add SEM, TEM, EDS, XRD, XPS. Specify the manufacturers of the materials and equipment used, as well as the modes used in the study of materials. For example: XRD scan angles, scan speed, geometric type and etc.

Response: According to your advices, the manufacturer of the TC4 titanium alloy specimens in this study is Shanghai Po Song Mechanical Electrical Equipment Company. The model of SEM is JSM-6390A, the manufacturer is Japan Electronics Corporation, and EDS is an additional module used for composition analysis. The model of XRD is Shimadzu XRD-6000, the scanning range is 5°-144° (2θ), and the power of X-ray tube is 2.7KW. XPS model is Thermo SCIENTIFIC ESCALAB Xi+, type: monochromatic Al target (E=1486.68 eV), Voltage: 14795.40 V, Current: 0.0108 A, Vacuum: P<10-9 mBar, Pass Energy: 100 eV ( Survey), 20 eV (High-resolutions), Work FN: 5.04 eV. The TEM model number is JEM-2100Plus and the manufacturer is Nippon Electron Corporation; the usage mode is elevated to 200 kV and the ion pump readings are less than 2X10-5 Pa. all of them have been added in paragraph 2.1. Please see Page 3.

  1. Line 123 Specify the exact type of reference electrode used.

Response: Thanks for your suggestions. The reference electrode used in this study was a PTFE Silver chloride electrode. Please see Page 4.

  1. Figure 8 - Expand the description of the phase composition. Give the ICDD numbers for each of the found phases.

Response: According to your advice, the phase of Al2O3 detected in XRD is Corundum, which belongs to α- Al2O3 and its PDF card number is 73-1512. The phase of TiO2 is Anatase, its PDF card number is 71-1166. And the phase of Ti is Titanium, its PDF card number is 88-2321. All of them have been added in the manuscript. Please see Page 8.

  1. Paragraph 3.1.2 it is not very clear how different mechanical loads affect the change in bond energy by the XPS method. Perhaps you have identified a different oxide composition of corrosion products depending on the load? Please present the results in a clearer way.

Response: The present study is to determine the valence forms of the elements by comparing the bond energies of the product elements under different conditions, and thus the corresponding oxides were determined.

  1. Paragraph 3.2.1 - It's not very clear why you have jumps of Ecorr and icorr. It is necessary to explain the observed results, rather than a superficial analysis, that the current density is a quantitative characteristic of the corrosion rate. How do mechanical loads affect corrosion performance and why?

Response: Thanks for your suggestions. As the corrosion current density is the ratio of corrosion current to the working surface area and the working surface area of the working electrode in this paper is 1 cm2, so the corrosion current density was showed in Table 7. Now the corrosion current has been adjusted to the corrosion current density, listed in Table 7.

When TC4 titanium alloy is subjected to elastic stress stretching, the absolute value of positive and negative external pressure increases, the chemical sites of the metal atoms excited by the deformation increases, the activity of TC4 titanium alloy increases, while the tensile stress also increases the grain spacing, there is a tendency to shift from the dense row to the non-dense row surface, providing energy for its corrosion behavior. TC4 titanium alloy after elastic deformation, by the thermal potential, chemical site changes lead to TC4 titanium alloy equilibrium potential and electrode potential shift in the negative direction, which means that the tendency of TC4 titanium alloy to be oxidized increases, and the tendency of metal autolysis also increases. Becasue the elastic deformation of TC4 titanium alloy electrode potential becomes negative, TC4 titanium alloy, as the anode of the corrosion cell, will be corroded, and the electrode potential becomes negative, so that the electric potential of the corrosion microcell increases, the circuit corrosion current increases, thus the corrosion of TC4 titanium alloy in the elastic stress state is accelerated.

  1. Paragraph 3.2.2 - The processing of the received data was not carried out correctly. Table 8 incorrectly indicates the units of measurement of the Warburg element and does not show the accuracy of χ2 data approximation. ‘’Cdl and Rct are the bilayer capacitance and charge transfer resistance’’ are not correct definitions. Cdl and Rct are the double electric layer capacitance and charge transfer resistance.

Response: Thanks for your suggestions, the presentation have been revised. Please see subsection 3.2.2 and Table 8 for details.

  1. Looking closely at the data in Table 8 and Graph 12 for the 40%σs sample, the table value for charge transfer resistance is 1569 Ohm cm2, while in Figure 12 this value is over 7000 Ohm cm2.

Response: The data has been checked revised. Please see Table 8.

  1. To be able to compare the results obtained by the EIS method and polarization curves, it is necessary to calculate the polarization resistance Rp and give a formula for calculation.

Response: The data for polarization resistance (Table 7 in Page 11) and the calculation formula (2) have been added in the manuscript in Page 5.

  1. Paragraph 4 - The mechanism of pitting corrosion has been described previously. Please add links to the authors who have studied these interactions. For example: https://doi.org/10.1016/j.ceramint.2021.12.318

Response: Thanks for your suggestions, the mechanism of pitting corrosion was revised according to the corresponding references. Please see Pages 14-16.

  1. It is also necessary to add possible chemical and electrochemical reactions to the pitting corrosion scheme.

Response: According to your advices, the possible chemical and electrochemical reactions to the pitting corrosion scheme have been added. Please see Pages 15-16.

  1. Usually, in the process of pitting corrosion on the polarization curves in the semi-logarithmic coordinates, there is a break in the anodic curve, which is characterized by pitting. You don't see this break. It feels like the process is accompanied by intergranular corrosion, and pitting corrosion is not observed here.

Response: As you said, the break in the anodic curve is not obvious, the pitting process may be accompanied by intergranular corrosion due to the different potential between α-phase and β-phase, and grains and grain boundaries.

Round 2

Reviewer 1 Report

The reviewer is not satisfied with the authors' answer to remark No. 30. The authors gave good answers to the comments of reviewer No. 1-29.

Reviewer 3 Report

Dear authors, I see major improvements of your manuscript. You answered clearly and satisfactory to all my review remarks. Your research paper can be published in the MDPI journal.